# Translocation of benzo(a)pyrene reactive metabolites across human mammary epithelial cell membranes

**B. Alex Merrick**[1]*, **Ashley M. Brooks**[2], **Julie F. Foley**[1], **Betty K. Mansfield**[3], **James K. Selkirk**[3]

**1** Mechanistic Toxicology Branch, Division of Translational Toxicology, National Institute of Environmental Health Sciences, Research Triangle Park, North Carolina, United States of America, **2** Integrative Bioinformatics Support Group, Biostatistics and Computational Biology Branch, Division of Intramural Research, National Institute of Environmental Health Sciences, Research Triangle Park, North Carolina, United States of America, **3** National Institute of Environmental Health Sciences, Research Triangle Park, North Carolina, United States of America

* merrick@niehs.nih.gov

## Abstract

DNA adducts from benzo(a)pyrene (BaP) and other polycyclic aromatic hydrocarbons (PAH) are related to tumor initiation in many tissues including mammary epithelia. T47D mammary cells are able metabolizers of BaP, forming DNA and cellular protein adducts but also a sizeable amount of extracellular protein adducts. GSH S-transferases (GST) help mitigate adduct formation by glutathione (GSH) conjugation. Here, we varied GSH levels using buthionine sulfoximine (BSO) and BaP pretreatments to deplete or augment GSH, respectively, to study adduct formation and metabolism at 4 µM $^3$H-BaP over 24–48hr. An inverse relationship was observed between GSH levels and nuclear protein and DNA adducts. Time course experiments showed extracellular protein adducts, identified primarily as bovine serum albumin and α1-AT (α-1-antitrypsin) in culture medium, were 5–10 times greater than cellular protein adducts and comprised 8–9% of total metabolized $^3$H-BaP. However, specific adduct binding (adducts/mg protein) in cells was much greater than for extracellular protein, likely from their intracellular proximity to CYP-mediated metabolism to BaP reactive metabolites. Proportions of $^3$H-BaP hydroxylated and conjugated metabolites in BSO and BaP pretreated cells were not greatly altered from DMSO control after 24 hr. Bioinformatic analysis of T47D cell gene expression indicated CYP1B1 and CYP1A1 were primary enzymes for BaP bioactivation. We surmised reactive BaP metabolites that escaped conjugation reactions were sufficiently stable to migrate into the

extracellular space. These results suggest BaP reactive metabolites like BPDE (BaP-diol-epoxide) can easily translocate across cell membranes despite robust conjugation systems and ready supplies of essential co-substrates for sulfate or GSH conjugations. The implications *in vivo* are that BaP reactive metabolites can enter adjacent epithelia and some fraction could result in DNA binding and somatic mutations in

**Data availability statement:** All relevant data are within the paper and its Supporting information files, and the RNA-seq data for gene expression analysis can be found in the Sequence Read Archives (SRA) database https://www.ncbi.nlm.nih.gov/sra. Fastq files from three published, independent RNA-seq studies can be found with the following Bioproject and fastq file identifiers. BioProject PRJNA920517 using GSM6921664, GSM6921665 and GSM6921666 files; BioProject PRJNA823675 using files GSM6022030, GSM6022031 and GSM6022032; and BioProject PRJNA910517 using GSM6807254 GSM6807255 and GSM6807256.

**Funding:** This research was internally funded by the Intramural Research Program of the NIH, National Institute of Environmental Health Sciences, through ZIA ES103378 from the Division of Translational Toxicology.

**Competing interests:** The authors have declared that no competing interests exist.

**Abbreviations:** AHR, arylhydrocarbon receptor; AHRR, arylhydrocarbon receptor repressor; ARNT, arylhydrocarbon nuclear receptor translocator; α-1AT, alpha-1-antitrypsin; BaP, benzo(a) pyrene; BPDE, (+)-anti-BaP-7,8 hydroxy 9,10 diol epoxide; BMC, base mean counts; BRCA, breast cancer gene; BSO, buthionine sulfoximine; CAT, catalase; CDNB, 1-chloro-2,4-dinitrobenzene; CYP, cytochrome P450; 2D-PAGE, two-dimensional polyacrylamide gel electrophoresis; 2D-gel, two dimensional gel; DMSO,dimethylsulfoxide; DPM, disintegrations per minute; EDTA, ethylene-diaminetetraacetic acid; Ensembl,–genome database at the European Bioinformatics Institute; EPHX, epoxide hydrolase; ER, estrogen receptor; GCLC, gamma-glutamylcysteine synthetase; GCLM, glutamate-cysteine ligase modifier; GPX, glutathione peroxidase; GSR, glutathione reductase; GSS, glutathione synthase; GST, glutathione S-transferase; GSTA, GST Alpha; GSTCD, GST C-Terminal Domain Containing; GSTK, GST Kappa; GSTM, GST Mu; GSTO, GST Omega; GSTP,GST Pi; GSTT, GST Theta; GSTZ, GST Zeta; HER2, human epidermal growth factor receptor 2; Hg38, human genome, version 38; KI67, marker of proliferation Kiel 67; LANCL, Lanthionine synthetase C-like protein;PAH, poly-cyclic aromatic hydrocarbons; PR, progesterone receptor;SOD, superoxide dismutase; SRA, Sequence Read Archives; SULT, sulfotransfer-ase; STAR, Spliced Transcripts Alignment to a Reference; TP53, tumor protein p53; UGT, UDP-glucuronosyltransferase.

cancer susceptibility genes over time. The relationship continues to grow between PAH exposure and pollution, and many malignancies including breast cancers.

## Introduction

Among reproductive diseases, breast cancer is a significant health concern for women worldwide. In the US, the incidence of breast cancer has continued to rise at 1% per year from 2012 to 2021 [1] with a disturbing trend of increasing incidence in young women [2,3]. The multifaceted etiology of breast cancer involves genetics, hormones, diet, lifestyle, and environmental factors. Genetically inherited mutations in BRCA1/2, TP53 and other genes [4–8] predispose carriers for up to 10% of all breast cancers [9,10]. Germline gene mutations leading to breast malignancies generally interfere with DNA repair, dysregulate cell cycle control pathways and alter chromatin remodeling [11–13]. Even so, somatic mutations frequently occur in breast cancer susceptibility genes. For example, mutations in the TP53 gene are most often somatically acquired [14] and up to one-third of BRCA1/2 mutations are somatic [15]. Genomic landscape surveys of breast cancer cohorts found BRCA1 and TP53 somatic mutations were more common in younger women at 20–40 years of age compared to older women at >65 years despite the increased mutational somatic load in older patients [16]. How somatic mutations arise in breast cancer susceptibility genes remains poorly understood. Environmental and dietary exposures can produce DNA mutational profiles that play an important role in breast cancer etiology [17,18].

Breast morphology is comprised of many steroid responsive epithelial structures including mammary glands, lobules and ducts, and invasive ductal carcinoma is the most common type of breast cancer [19]. Subtypes are immunohistologically categorized as luminal Type-A (ER/PR+, HER2-) or Type-B (ER/PR+, HER2-, Ki67+). Cell lines are an invaluable source for testing new chemotherapeutics, analyzing gene function, and investigating breast epithelial cell biology. T47D cells are a human mammary epithelial tumor cell line originally isolated from pleural effusion fluid [20]. They exhibit a cobblestone appearance in culture and display immunohistological markers indicating a luminal Type-A origin [21]. T47D cells are capable of metabolizing benzo(a)pyrene (BaP), a representative environmental carcinogen, to form DNA adducts [22–24].

Evidence of environmental contributions to mammary mutational burden and gene-specific damage leading to breast cancer continues to accumulate [8,25,26]. Polycyclic aromatic hydrocarbons (PAH) remain a major risk factor in environmental health [27]. PAH exposure occurs through multiple sources including natural (e.g., wildfire smoke, food, cooking), industrial sources (e.g., fossil fuels), and lifestyle choices (e.g., smoking) of which benzo(a)pyrene is representative of this class [27–30]. BaP metabolism has been extensively studied in human cell cultures, producing hydroxylated metabolites, and conjugates [31]. Although BaP metabolites can bind to many cellular macromolecules, DNA and protein adducts

are the most biologically consequential [32]. DNA adduct formation varies according a cell type's relative capacity to generate electrophilic metabolites such as (+)-anti-BaP-7,8 hydroxy- 9,10 diol epoxide (BPDE) balanced with conjugation reactions to produce inactive forms, primarily by the conjugating activity of glutathione S-transferase isoforms (S1 Fig) [31]. Prior work by us and colleagues, reported BaP activation by T47D cells results in DNA and protein adducts as well as hydroxylated and conjugated metabolites [33,34]. The glutathione S-transferase (GST) family is comprised of 7 family members with many overlapping functions but some have specificity for PAH detoxification including BaP metabolites [35]. For example, transfection of specific GST isoforms such as GSTP1 (GST-Pi) into T47D cells reduced BPDE cytotoxicity and increased GSH conjugates [22]. Further, altering cellular GSH levels by buthionine sulfoximine (BSO), an inhibitor of γ-glutamyl cysteine synthetase, provides another tool to study cellular mechanisms of adduct formation. BSO-mediated depletion of GSH sensitized T47D and other breast cancer cell lines to cytotoxicity by the alkylating agent, hepsulfam, since GSH is required for its enzymatic detoxification [36]. Pretreatment with BaP in human cell cultures reportedly increases GST levels by AHR activation [37,38], but this has yet to be tested in T47D cells. T47D cells have adequate GST activity for most conjugation reactions [36,39], although the exact distribution of GST family members is currently not known for this cell line.

Our prior work suggested the distribution of BaP adducts has been greatly underestimated in cell culture models and that this observation has relevance for *in vivo* breast cancer risk [33]. We believe the relative stability of BaP electrophiles (e.g., BPDE), after generation in mammary cells, plays an important role in affecting adjacent cells and distal targets beyond those cells responsible for metabolic activation [40,41].

In the current study, we were interested in the effects of altered GSH levels by BSO or BaP pretreatments upon ³H-BaP metabolite and adduct distribution in T47D breast epithelial cells. We hypothesized an inverse distribution of cellular DNA/protein adducts and extracellular protein BaP adducts would be found after GSH depletion or augmentation in T47D cells.

## Materials and methods

### Cell culture

T47D cells were obtained from the ATCC (Cat No. HTB-133). Cells were grown in 100 mm culture dishes in 10 ml of RPMI 1640 media supplemented with 10% fetal bovine serum, L-glutamine at 2mM, penicillin-streptomycin and insulin at 10 μg/ml, all of which were supplied by Gibco, Life Technologies. Cells were propagated at 95%:5% air and $CO_2$ at 37°C. Cells were grown to 80–90% confluency prior to chemical treatment.

### Chemical sources

Chemicals were obtained from the following sources: unlabeled BaP was obtained from Calbiochem-Millipore (San Diego, CA); ³H-BaP (4μM, 6 Ci/mmol) was from Amersham Biosciences (Chicago, IL); buthionine sulfoximine, dimethylsufoxide (DMSO), metaphosphoric acid, EDTA, guanidine-HCl, cesium sulfate, 1-chloro-2,4-dinitrobenzene (CDNB) and reduced glutathione (GSH), were from Sigma-Aldrich (St Louis, MO).

### GSH measurement

Cellular GSH was measured enzymatically due to assay specificity for GSH without interference from other cellular sulfhydryl molecules [42]. Briefly, cells were scraped from 100 mm culture dishes, pelleted at 4°C and homogenized in 10% meta-phosphoric acid. Precipitated protein was centrifuged at 4°C in a microfuge. Supernatant was analyzed for GSH in a volume of 3 ml potassium phosphate buffer (pH 6.5) with 0.6 μM CDNB and 0.8 IU of GSH S-transferase activity. Experimental samples and GSH standards (up to 200 nmoles) were incubated at 37°C for 10 min and then read in a UV-spectrophotometer at 340nm.

## BaP adduct measurements

$^3$H-BaP adducts were measured from protein in culture medium and from T47D whole cellular protein at various time points [33]. Briefly, 10 ml of culture media protein from a 100 mm dish was removed into a separate tube and then precipitated with acetone. Precipitated protein was pelleted and washed 10 times with an 80% methanol aqueous solution. This procedure removed non-specific BaP binding prior to protein dissolution, measurement of radioactivity and protein content. Similarly, cells matched with culture media adducts, were scraped from one 100 mm dish, pelleted, and 80% methanol-washed 10 times before protein was dissolved for radioactivity counting and protein analysis. All groups were background corrected for non-specific binding. The term "specific adduct binding" refers to amount of adducts specific per mg protein (e.g., pmole BaP adducts/mg protein) or per mg DNA, so as to normalize adduct formation according to protein levels in cells or culture media (extracellular protein) or DNA levels.

Nuclear protein and DNA adducts were determined as previously described [43]. Briefly, T47D cultures were incubated with 4 µM $^3$H-BaP for 24 hr and then cells were scraped, pelleted and lysed in Triton X-100 and sodium deoxycholate at 0.5% by glass mortar-pestle homogenization. Nuclei were pelleted for dissolution in guanidine-EDTA, unbound BaP was removed, and lysates were layered over an isopycnic cesium solution for centrifugal isolation of DNA and protein. DNA was quantitated in drop aliquots by spectrophotometry; protein was measured by fluorescamine; and radioactivity was measured by scintillation counting.

## Distribution of BaP metabolites

Incubation of T47D cells with 4µM $^3$H-BaP was carried out for 24hr. Media and cells were collected to determine the distribution of BaP metabolites as previously described [33]. Briefly, media from five 100 mm dishes (total of 50 ml) was first extracted with ethyl acetate to measure the amount of free hydroxylated metabolites. Remaining media was then enzymatically digested with β-glucuronidase/arylsulfatase (Sigma Cat No. BGALA-RO) and re-extracted with ethyl acetate to determine the amount of conjugated hydroxylated BaP metabolites. Culture media protein was precipitated with acetone and the residual media was measured as $^3$H-BaP water-soluble metabolites. The same set of procedures was used to extract and count $^3$H-BaP metabolites from T47D cells as free, conjugated and water-soluble metabolites. The amount of protein adducts were quantitated by measurement of radioactivity and protein content by fluorescamine.

## 2D-Gel electrophoresis

Separation of T47D cellular or culture media proteins was conducted by two dimensional polyacrylamide gel electrophoresis (2D-PAGE) to determine $^3$H-BaP adducts. These protein separation methods have been reported [44]. To summarize, T47D cellular or culture media proteins were dialyzed, lyophilized and solubilized in a urea lysis buffer containing pH 4–8 ampholytes. Proteins were separated by isoelectric focusing and then by size in polyacrylamide slab gels in a 10% to 16% linear gradient. Radiolabeled proteins were fixed in an acetic acid-ethanol solution and detected after 1 or 2 months by film fluorography at −80°C.

## Gene expression analysis

Gene expression of untreated, resting T47D human breast cells was determined for assessment of drug metabolism and detoxification capabilities of this cell line. Fastq files from three different, independently conducted RNA-seq studies were downloaded from the NCBI SRA (Sequence Read Archives) database. Requirements were that these RNA-seq studies were recently published, involved paired-end sequencing on independent triplicate samples on an Illumina instrument platform, in which such samples served as untreated, controls in the experiment. BioProject numbers and file identifiers can be found in the Data availability statement. Reads from the three RNA-seq studies were aligned to hg38 with the open source software packages, STAR (Spliced Transcripts Alignment to a Reference) v.020201 and were normalized with

the DESeq2 v.1.44.0 median of ratios method for each gene. Genes were annotated by gene symbol, Ensembl ID, gene name, RefSeq and EntrezID identifiers, and protein-coding type. The base mean count (BMC) was calculated for each Ensembl gene. Heat maps were constructed using Partek Genomics Suite version 7.0 (Partek/Illumina, San Diego, CA) for CYP and detoxification-related genes (≥5 BMCs for one or more study groups) related to BaP metabolism.

### Statistical analysis

Multiple treatment groups for DNA/nuclear protein adducts, and GST activities at a fixed time were analyzed by one-way ANOVA and Newman-Keuls post-hoc testing at $p < 0.05$ significance. When measurements were analyzed over multiple treatments and times points for GSH and protein adducts, two-way ANOVA and Newman-Keuls post-hoc tests were performed at $p < 0.05$ significance.

## Results

### Time course of GSH levels after BSO or BaP treatment

Cellular GSH levels are an important determinant in conjugating BaP electrophiles. Diminished GSH levels could allow increased formation of BaP adducts. We investigated conditions to raise or lower T47D cellular GSH levels prior to examining BaP adduct formation. Initial experiments were designed to decrease GSH levels with an inhibitor of GSH synthesis, buthionine sulfoximine (BSO). In addition, we tested the ability of 4 µM BaP (unlabeled) to increase cellular GSH.

Results in Fig 1 Panel A show BSO caused a gradual drop in cellular GSH levels which stabilized at about 10% of initial levels by 48 hr. When cells were exposed to 4 µM BaP, GSH levels rose rapidly after 6 hr and peaked by 24 hr at about 60% above DMSO control. GSH levels were maintained at >50% above DMSO control after BaP treatment and declined only slightly over 72 hr. GSH levels in control cells treated with DMSO rose slightly (7% maximum) which was probably related to the effects of providing fresh media at 0 hr. These results suggested that a 48 hr pretreatment was an effective period for lowering or raising GSH levels with BSO and BaP treatments, respectively.

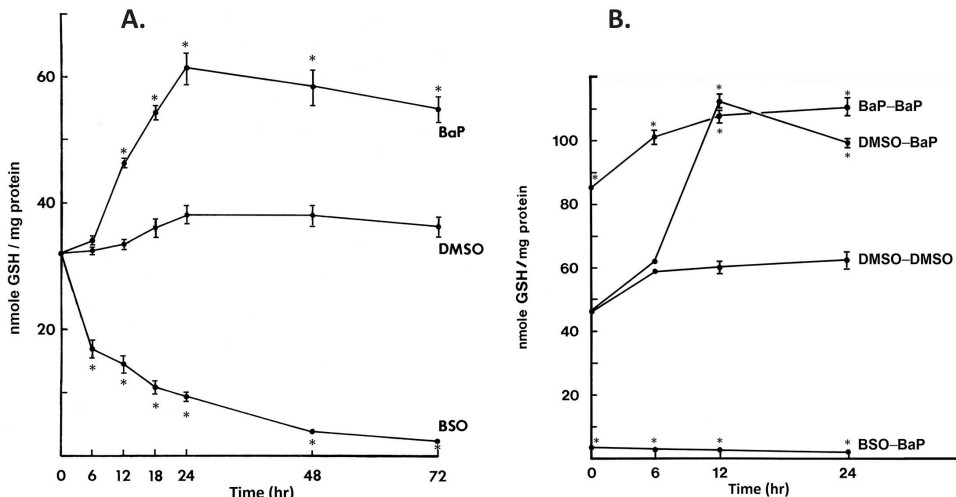

**Fig 1. GSH levels in T47D cells after BSO or BaP.** (A) Panel A shows changes in T47D cellular GSH levels after treatment with 4 µM BaP or 0.5 mM BSO. DMSO at 0.4% v/v was the control. Values represent the mean ± S.E. at n = 3/group. **(B)** Panel B shows cellular GSH levels in cells pretreated for 48 hr with DMSO (0.4% v/v), BSO (0.5 mM) or BaP (unlabeled, 4 µM) and then responding to a subsequent treatment with 4 µM BaP. Here, control T47D cells were only treated with DMSO. Values represent the mean ± **S.**E. at n = 3/group. Data in each panel were analyzed by two-way ANOVA over time and treatment. Post-hoc testing indicates $p < 0.05$ significant difference (asterisk) of treatments from the control time point.

Prior to using radiolabeled BaP for adduct formation, we wanted to know what effects BaP would have on GSH levels in T47D cells pretreated 48hr with DMSO, BSO and BaP. Time course results over 24hr after BaP treatment are shown in Fig 1 Panel B. Cells pretreated with DMSO and BaP for 48hr reached GSH levels that were over twice the amount of controls receiving no pretreatment. DMSO pretreated cells experienced a rapid rise in GSH after 6hr which was maintained at 24hr after BaP exposure. BaP pretreated cells exposed to BaP a second time (already at 80% above DMSO GSH levels at 0hr) experienced a gradual increase in cellular GSH levels. BSO treated cells continued to exhibit depressed cellular GSH levels at 5% above GSH of DMSO control levels throughout the 24hr treatment period to BaP. Therefore, BSO pretreated cells were resistant to the inductive effects of BaP exposure while DMSO, and BaP pretreated cells showed induced levels of cellular GSH. In summary, a period of 48hr treatment with BSO or BaP was found to substantially decrease (<5%) and increase (>80%) cellular GSH levels in T47D cells, respectively from DMSO control.

## $^3$H-BaP adduct formation after BSO or BaP treatment over time

Based on the prior experiments, we examined the kinetics of $^3$H-BaP adduct formation under conditions that depleted or increased GSH levels. We tested if there was an inverse relationship between cellular GSH concentrations and cellular $^3$H-BaP adduct levels. Cellular and extracellular adducts were measured by acetone precipitation of protein, exhaustive washing with MeOH:H$_2$O and radiometric counting of solubilized protein. Fig 2 Panel A shows the rapid rise of cellular protein adducts normalized per mg protein in GSH depleted cells treated with BSO, reaching a maximum after 18hr at 89% above DMSO control. Induction of GSH levels by BaP pretreatment lowered the formation of cellular $^3$H-BaP adducts by 20–25% at 12–24hr compared to DMSO control. Thus, an inverse relationship was observed between cellular GSH and BaP cellular adduct formation, although to a greater magnitude in BSO pretreated cells *versus* BaP pretreatment.

We did notice the specific adduct binding levels of cell protein in the BSO group started to decline at 24 and 48hr compared to the peak at 18hr (Fig 2 Panel A). Similarly, specific adduct binding levels in the DMSO group also began to decline by 24 and 48hr from their peak values at 12 and 18hr - eventually merging with BaP group adduct levels at 48hr. To explain these observations, we reasoned BSO and DMSO treated cells had continued cell growth (Fig 2 Panel B) thereby decreasing their specific adduct binding levels. By contrast, the BaP treated group experienced reduced growth, likely from inhibitory effects of a double BaP exposure (BaP 48hr pretreatment, followed by $^3$H-BaP exposure) so that this group's specific adduct binding was not as greatly affected by cell growth.

Panel C of Fig 2 shows accumulation of extracellular $^3$H-BaP adducts over time for each group. Unlike cellular protein, extracellular protein content was constant at 3mg/ml. Over 48hr, DMSO and BaP pretreated groups essentially accumulated the same extracellular adduct levels (except at 6hr). By contrast, specific adduct binding of extracellular protein in BSO samples were significantly lower from18–48hr compared to the other two groups. The formation of extracellular adducts began to diverge after 12hr of radiolabel treatment. By 48hr, DMSO and BaP groups had 19% and 14% more extracellular adducts, respectively, than BSO. Even so, extracellular adducts normalized per mg media protein showed a continued climb for all treatment groups over a 48hr period.

Another observation was that the specific adduct binding in cells shown in Fig 2 Panel A was always greater than specific adduct binding in extracellular media protein in Fig 2, Panel C (note y-axis scales in panels A and C). This observation was consistent for all treatment groups at all time points from 6 to 48hr. The magnitude of specific cellular vs extracellular adduct binding was most pronounced in BSO pretreated cells (~10-fold higher from 6 to 18hr). DMSO and BaP pretreated cells showed increases in cellular vs extracellular specific adduct binding of 2-fold to 5-fold over the 48hr time course. Notably, the amount of cellular protein available for BaP binding (1.1 to 1.7mg over 24hr) was almost 10-fold less than extracellular media protein measured at a constant 15mg total (3mg/ml x 5ml volume). No cytotoxicity occurred at 4 µM BaP under these incubation conditions.

The kinetics of total cumulative BaP adducts (pmole cellular + pmole extracellular adducts) in Fig 2 Panel D indicates that overall, $^3$H-BaP binds total protein more rapidly in BSO treated cells compared to DMSO control

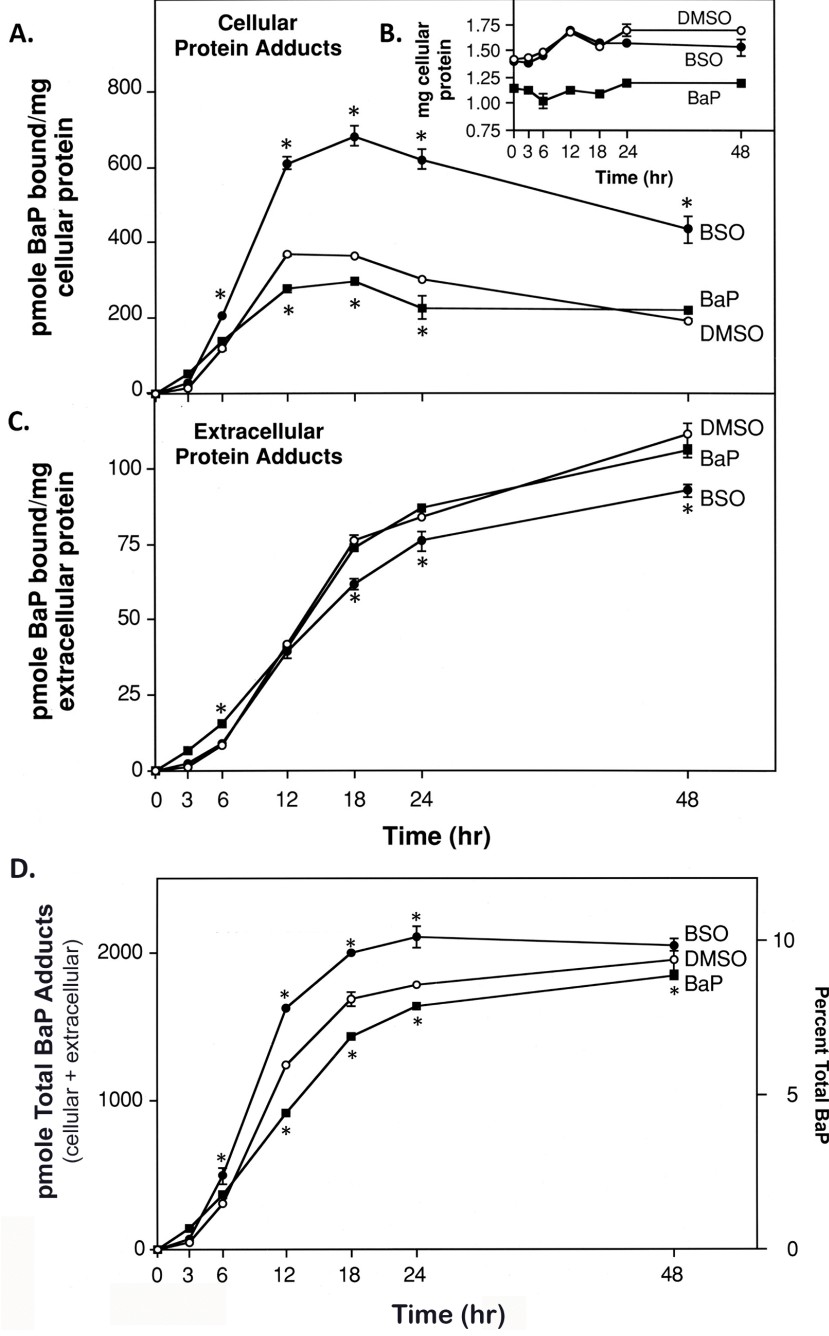

**Fig 2. BaP protein adduct formation over time in T47D cells.** All line graphs in Panels A-D indicate pretreatment of cells for 48hr with DMSO (0.4% v/v control), BSO (0.5mM), or BaP (unlabeled, 4 μM) prior to exposure to $^3$H-BaP to measure adduct formation. **(A)** In Panel A, BaP cellular protein adducts are shown over 48hr. **(B)** Panel B insert displays cellular protein concentrations which were used to normalize cellular adduct levels at each time point. **(C)** Panel C shows extracellular media protein adducts over 48hr, normalized to media protein content. **(D)** Panel D represents the sum total of protein adducts (cellular + extracellular protein adducts) accumulated at each time point. Each time point value represents the mean ± S.E. at n = 3/group. Data were analyzed by two-way ANOVA over time and treatment with post-hoc significance testing at p < 0.05 (asterisk) of BSO or BaP treatments compared to the DMSO control time point. Please see text for further details.

during the initial 24 hr of incubation, while BaP pretreatment produced less adducts than the DMSO control. These results are consistent with a hypothesized inverse effect of GSH levels on BaP adduct formation. Interestingly, the cumulative total level of all $^3$H-BaP adducts at 48 hr in all treatment groups approached a similar value at 9–10% of total BaP added per dish when all free BaP had been metabolized. Total pmole BaP adducts at 48 hr for BSO (2,050.2 ± 40.4) and for BaP (1842.3 ± 38.0) pretreated groups were within ±5% of DMSO controls (1,947.1 ± 66.1). Unlike in *vivo* conditions where metabolites are removed by blood flow, we speculate that either an enzymatic or spontaneous decoupling of some BaP conjugates *in vitro* may lead to continued metabolism, contributing to increased total protein binding. Additional studies are needed to clarify this finding. Data in Fig 2 demonstrate the kinetics of protein adduct formation is primarily influenced by 1) the amount of cellular vs extracellular protein available for binding and 2) the proximity to reactive metabolite formation within the cell. In Fig 2 Panel A, amounts of cellular adducts were normalized by approximately 1.5 mg of cellular protein. The data show a high level of adducts being formed as pmole adducts/mg cellular protein, especially in the BSO group where GSH is reduced compared to DMSO control. In Fig 2 Panel C, the larger amount of 15 mg extracellular protein available for binding shows a somewhat lower level of adducts formed in the BSO group vs DMSO control, expressed as pmole adducts/mg extracellular protein. This may be due to cellular protein capturing more reactive BaP metabolites under low GSH levels with BSO, that would otherwise migrate out of the cell and react with extracellular protein. Fig 2 Panel D shows the kinetic perspective for the total amount of pmole BaP adducts formed as the sum of cellular and extracellular adducts at each time point, where the total amount of protein available for BaP binding is similar for all groups (~1.5 mg cellular protein + 15 mg extracellular protein = 16.5 mg total protein). From this view, the BSO group has the highest total amount of adducts formed, likely due to the reduction of GSH for conjugation, leaving more reactive BaP reactive metabolites to create adducts compared to DMSO control. For the BaP pretreatment group, the increase in cellular GSH for conjugation reactions resulted in a lower amount of total protein adducts formed compared to DMSO control.

The effects of GSH depletion or induction upon BaP adduct formation were also determined in T47D cell nuclei. After near depletion of GSH with BSO (Table 1), DNA adducts/mg increased by 29% after 24 hr while nuclear protein adducts/mg almost doubled. Increasing GSH by BaP pretreatment did significantly lower DNA adducts/mg by 13% but did not significantly affect nuclear protein adducts/mg. Interestingly, when protein BaP adduct formation from whole cell lysates was followed over time (Fig 2, Panel A), it generally followed the same proportional binding pattern as nuclear protein adducts for BSO treatment. At 18 and 24 hr, specific protein adduct binding in cells after GSH depletion was almost doubled from the DMSO control. However, GSH increases from BaP pretreatment lowered cellular adducts in Fig 2, Panel A by relatively lesser amounts to 20–25% over 12–24 hr from DMSO control, while nuclear protein binding with BaP pretreatment (Table 1) was not statistically lowered from control. Apparently, the increase in GSH caused by BaP pretreatment was not substantial enough to lower nuclear protein binding.

**Table 1. BaP adduct levels in nuclear protein and DNA of T47D cells.**

| Pretreatment | Nuclear protein | DNA |
|---|---|---|
| Specific adduct binding (pmole BaP adducts/mg macromolecules) | | |
| DMSO | 92.2 ± 9.3 (100%) | 84.4 ± 4.2 (100%) |
| BSO | 178.6* ± 8.2 (194%) | 108.8* ± 2.1 (129%) |
| BaP | 101.9 ± 12.4 (111%) | 73.0* ± 3.4 (87%) |

Confluent T47D cell cultures were treated for 48 hr with DMSO (0.4% v/v), 0.5 mM BSO or 4 µM BaP prior to receiving $^3$H-BaP at 4 µM for 24 hr. Values represent the mean ± S.E. at n = 3/group. An asterisk symbol indicates significant difference from DMSO control (p < 0.05 by ANOVA, Newman-Keuls post-hoc testing).

## T47D metabolism of BaP during GSH depletion or induction

We previously reported that T47D cultures metabolize ≥95% of BaP during a 24 hr incubation period in which sulfation of metabolites was the predominant conjugate compared to glucuronide conjugation [33]. Our hypothesis in this study was that changing GSH levels with BSO or BaP would substantially alter the distribution of ³H-BaP metabolites and adducts. After a 24hr incubation period with 4 μM ³H-BaP, less than 2% BaP remained, indicating metabolism was essentially complete. In our analysis, we accounted for metabolites and adducts in both cells and media. Data are summarized in the circle chart of Fig 3 (S1 Table). Results show that the proportions of hydroxylated metabolite (shown as free unconjugated metabolites – blue bands) or as conjugated metabolites (grey bands) were greatest in culture media compared to cellular metabolites, similar to our earlier work. Free hydroxylated BaP metabolites were at 18.6% and 17.3% in DMSO and BSO groups, respectively, and slightly higher at 22.6% in the BaP pretreated group. Comparable metabolite levels at 29–30% were seen in conjugated hydroxylated metabolites in media among the three treatment groups. Only a small amount of free hydroxylated metabolites could be found in T47D cells at 2–3% of total, while no conjugated metabolites in cells were observed in the three treatment groups. The proportions of water-soluble metabolites (yellow bands) were those remaining after sulfatase/glucuronidase digestion and were generally comparable in media among treatment groups (36.1%, 35.7% and 34.7% for DMSO, BSO and BaP, respectively) or in cells (1–2%).

As might be expected, the combined amount of conjugated or unconjugated BaP metabolites, when compared to total protein adducts in Fig 3, was much greater at a ratio of 10–1. However, there were revealing differences in adduct distributions among treatment groups, especially in cellular protein adducts. The 1% proportion of cellular adducts in the DMSO group was almost doubled in the BSO pretreated group at 1.9% where GSH was significantly reduced, while cellular adducts were lowered by about one-third to 0.7% in the BaP pretreatment group where GSH levels were increased. Even so, the proportion of extracellular protein adducts found in culture medium in BSO and BaP pretreated groups at 7.7% and 7.8%, respectively, was not substantially different from the DMSO control group at 8.6%. Secondly, these observations also suggest a substantial escape of reactive BaP intermediates into culture media occurred, despite considerable metabolite conjugation processes in T47D cells. We measured the GSH S-transferase (GST) enzyme activity in T47D cells among the 3 pretreatment groups (Table 2) that showed statistically similar levels among all groups, noting there were slight increases in activity in BSO and BaP groups compared to DMSO control.

## Electrophoresis separation of protein adducts

Separation of ³H-BaP media protein adducts was performed by 2D PAGE (two-dimensional polyacrylamide gel electrophoresis) after culture media was concentrated, frozen, lyophilized and dissolved in urea-lysis buffer for electrophoresis. 2D PAGE analysis images of extracellular protein adducts found in culture medium are shown in Fig 4. The two primary adducts at 66 kD and 60 kD are visible in the lower panel and correspond to principal components of bovine serum albumin and α1AT (alpha-1-antitrypsin) respectively, as identified by electrophoresis and mass spectrometry in our lab, and others [45–47]. Additional minor adducts at 72 kD, 62 kD, 53 kD, 52 kD and 26 kD were also observed but these have not yet been identified. Such minor adducts could be cellular proteins leaking out from cell membranes. While possible, the maintenance of cell viability and intact membrane morphology throughout the treatment period make this possibility unlikely. Similar electrophoretic results were obtained from media collected from BSO or BaP pretreated samples. We also attempted to detect additional extracellular protein adducts by application of larger amounts of radioactivity. However, application of extracellular protein samples at >50,000 dpm (disintegrations per minute) resulted in gel failure at the isoelectric focusing stage and unacceptable protein resolution.

Cellular ³H-BaP protein adducts were also separated by 2D PAGE (Fig 5). We first compared separation of protein cellular ³H-BaP adducts to relative amounts of protein by silver staining in DMSO controls. The fluorographic intensity of adduct proteins is clearly different from the silver staining pattern of cellular proteins, suggesting adduct formation was not correlated to the relative proportions of individual proteins. We then compared the pattern of ³H-BaP cellular adducts separated from cells exposed to DMSO, BSO and BaP pretreated cells. Gels were each loaded with 500,000 dpm for 2D

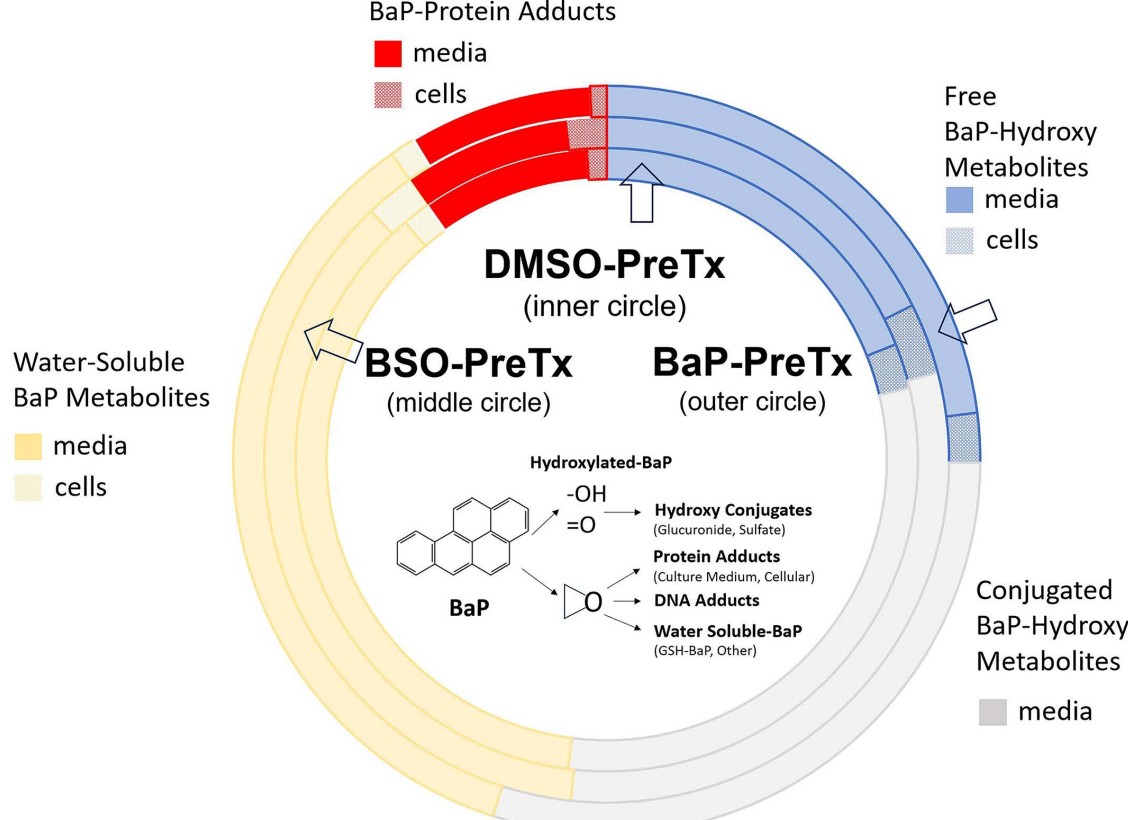

**Fig 3. Distribution of BaP metabolites and adducts in T47D cells.** Cells were incubated with [3]H-BaP (6 Ci/mmole) at 4µM for 24 hr. BaP was almost completely metabolized with <2% remaining at the end of incubation. The distribution of free hydroxylated metabolites (unconjugated) and sulfate/glucuronide conjugated metabolites after enzymatic digestion were determined by HPLC. Water-soluble metabolites were measured after enzymatic treatment with sulfatase/glucuronidase. Cellular adducts and extracellular adducts in media were measured by scintillation counting after protein precipitation, extensive washing and solubilization. Data represent the average of two independent metabolism experiments. See Methods for further details.

**Table 2. GSH S-Transferase activity in T47D cells.**

| Treatment | Specific enzyme activity (nmole product/min/mg protein) |
|---|---|
| DMSO | 50.3±2.1 (100%) |
| BSO | 59.3±1.4 (118%) |
| BaP | 61.5±4.1 (122%) |

Confluent cultures of T47D cells were treated for 48 hr with DMSO, 0.5 mM BSO or 4 µM BaP. GSH S-transferase activity was determined from 100,000 x g supernatant using chlorodinitrobenzene (CDNB) as a substrate. Results are the mean ± S.E. at n=3/group. ANOVA showed activities were not statistically different among groups.

PAGE separation. S2 Fig shows that although there were some minor differences, the electrophoretic separation patterns were similar among treatment groups. These data suggest that GSH depletion or induction may increase or decrease the amounts of specific protein adducts formed in cells but did not greatly influence those specific target proteins bound by reactive [3]H-BaP metabolites.

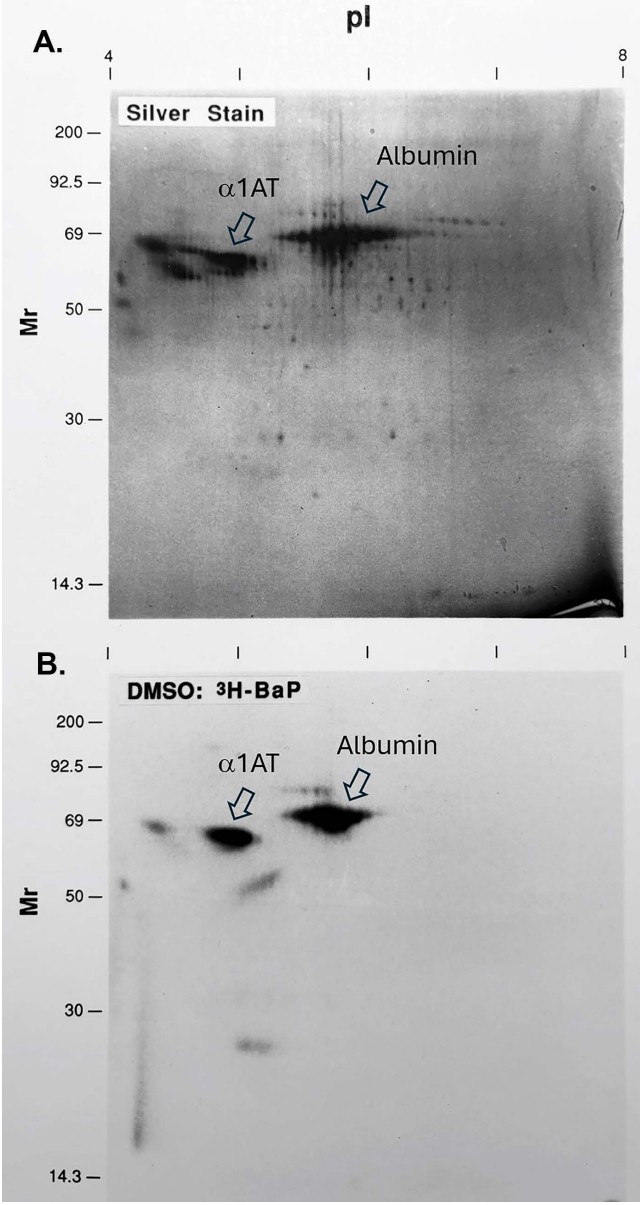

**Fig 4. BaP extracellular protein adducts in culture media after separation by 2D-PAGE.** Culture media protein was concentrated and separated by 2D gel electrophoresis. The sample shown was from media of DMSO pretreated T47D cells, treated with 4 μM ³H-BaP for 24 hr. **(A)** Detection of extracellular media protein was by silver gel staining (top panel) and **(B)** by fluorography of a 50,000 dpm load that was electrophoresed, gel-dried, and then exposed to film for 4 weeks (bottom panel). Two major ³H-BaP protein adducts were albumin and alpha-1-antitrypsin (α-1AT). BSO and BaP pretreated culture media produced the same protein separation pattern.

Unlike the few abundant culture media proteins that limited radioactivity loads to 50,000 dpm for acceptable 2D gel resolution, the mass and isoelectric focusing properties of cellular proteins were more widely distributed and allowed for increased amounts of labeled cellular protein to be loaded and resolved. Approximately 110 cellular protein adducts were detectable under current analytical conditions. A load amount of 500,000 dpm ³H-BaP bound cellular protein was a practical upper limit for consistent isoelectric focusing resolution above which the general radioactivity background increased without raising the number detectable protein adducts by fluorography.

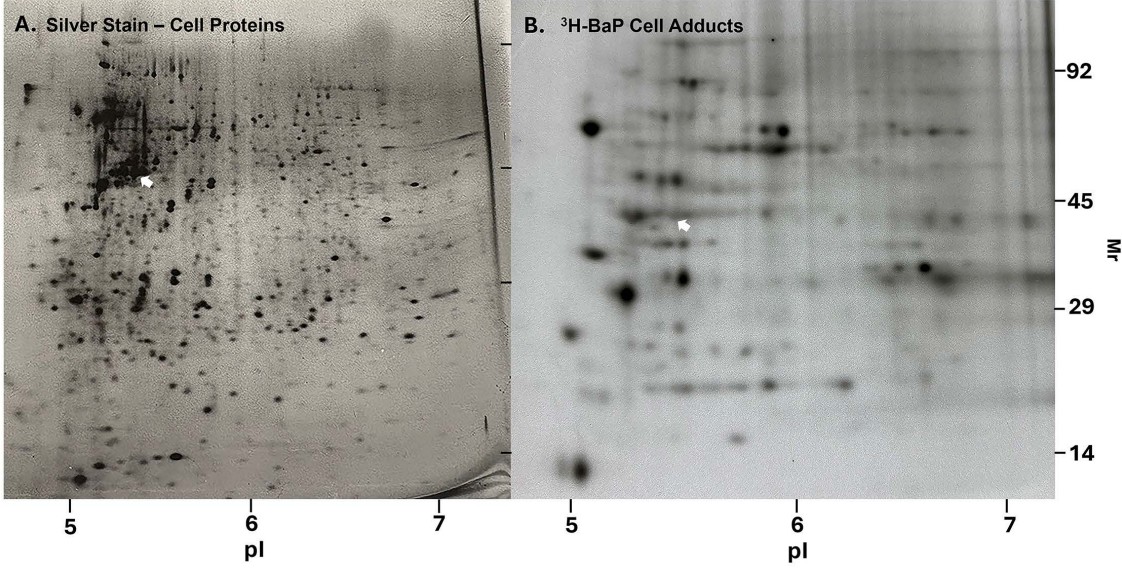

**Fig 5. Silver staining and ³H-BaP cellular protein adducts in T47D cells.** Cellular protein from T47D cells (20 µg) was separated by 2D PAGE. **(A)** Gels were silver stained and photographed. **(B)** In a separate experiment, T47D cells were exposed to 4 µM ³H-BaP for 24 hr after which cellular protein was precipitated, washed, lyophilized and solubilized for electrophoresis. A 500,000 dpm amount of labeled protein was separated by 2D PAGE, after which gels were fixed, dried and exposed to film for 8 weeks at −80°C. β-actin was determined by co-electrophoresis of western blots to fluorographs.

## Bioinformatic query of CYP metabolism and detoxification genes in T47D cells

Transcript profiling for metabolic and detoxification capabilities of T47D cells have not yet been described. Although we did not perform transcriptomic analysis, others have reported RNA-seq data on untreated control T47D cultures that provide insight into this cell line's potential for BaP metabolism [48–50]. Study samples were named either 'Untreated', 'Normoxia' and 'Parent' since we wanted to use the same nomenclature for the control groups that researchers designated in each particular study. RNA-seq reads were aligned by us to 60,649 Ensembl genes in hg38 and were normalized and expressed as base mean counts (BMC) to indicate the relative number of reads among all genes (S2 Table).

Heatmaps shown in Fig 6 focus only on expressed CYPs and detoxification genes relevant to BaP metabolism. Gene expression for CYPs and detoxification genes in these three studies showed consistency, especially in highly and moderately expressed genes. The high expression of CYP1B1 is important for metabolism of BaP to reactive metabolites along with mild expression of CYP1A1, and minimal expression of CYP1A2. These three CYP enzymes are a primary focus of cellular bioactivation of BaP, although low expression of CYP2D6 and CYP2C9 may also a play minor roles [30]. Detoxification of BaP relies upon conjugation reactions of which SULT1A1 and SULT2B1 (sulfotransferase) for sulfation are well expressed in T47D cells, while mild expression occurred for UGT3A2 (UDP-glucuronyltransferase) as the only enzyme for glucuronidation. GSH synthesis enzymes were sufficiently expressed to provide GSH for transferase reactions to conjugate BaP epoxides and diol-epoxides. Expression of GSH S-transferases is cell-type dependent and each isoform may have overlapping or specific affinities for BaP electrophilic substrates, especially BaP diol epoxides such as the GSTM (GST-Mu) isoform series. GSTM3 is the most highly expressed enzyme while GSTM2 and GSTM4 are moderately expressed. GSTM1, with high affinity for BaP diol epoxides, is not expressed. Several other GST isoforms are moderately expressed, including GSTO1, GSTO2, GSTK1 and GSTZ1 (GST-Omega, GST-Kappa, GST-Zeta, respectively). Other enzymes that conjugate secondary reactive oxygen species such as GSH peroxidases (GPX3, GPX4, GPX8), superoxide dismutases (SOD1, SOD3), and epoxide hydrolases (EPHX1, EPHX2) are highly or moderately expressed. Also important

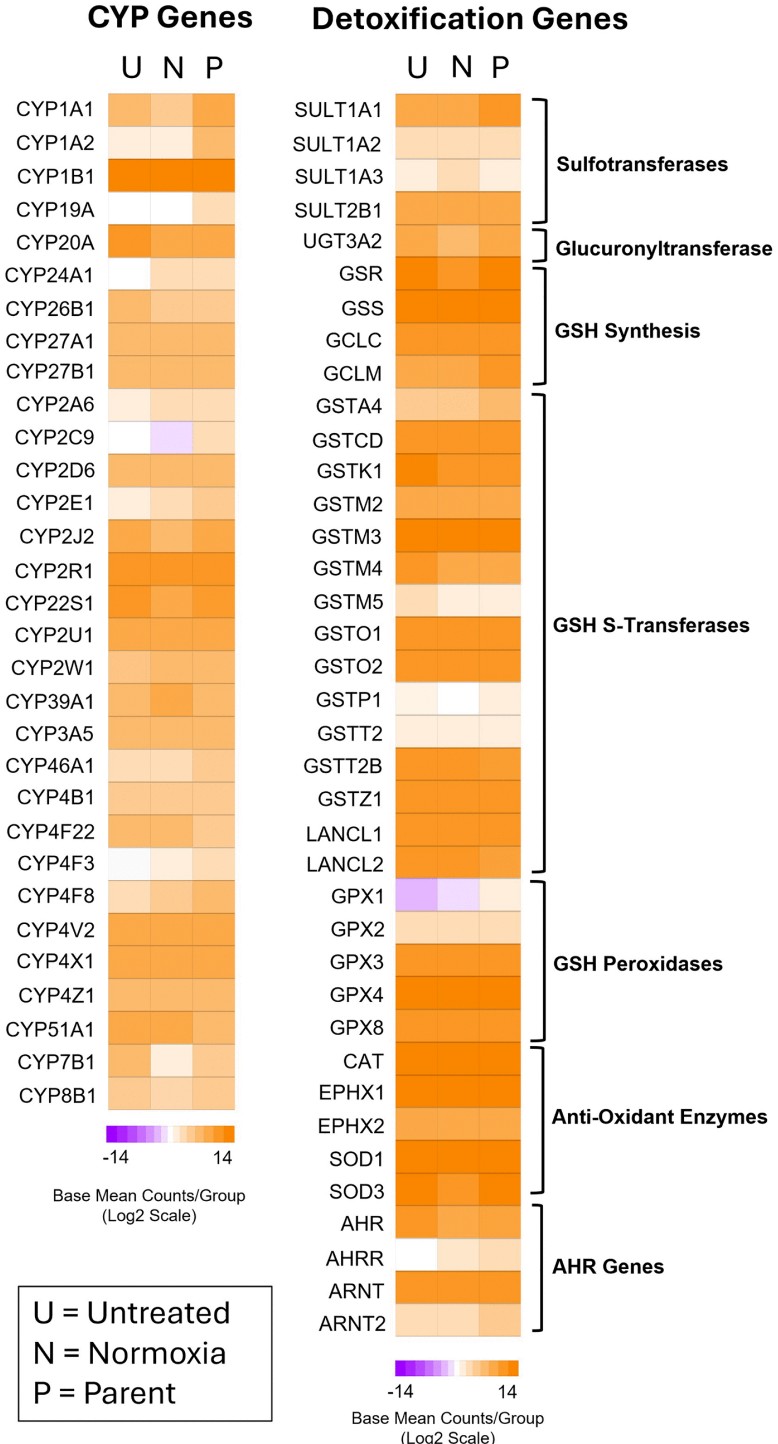

U = Untreated
N = Normoxia
P = Parent

**Fig 6. T47D metabolism and detoxification capabilities.** RNA-seq files were downloaded from 3 different published studies in the SRA database. Each study contained RNA-seq datafiles for triplicate, independent T47D cell cultures that served as controls. Data were designated as Untreated, Normoxia and Parent in accordance with each study's labeling for untreated control cultures. RNA-seq reads were aligned to Ensembl genes in hg38, and were normalized and expressed as base mean counts (BMC). BMCs were averaged for each transcript and converted to log2 scale. Only genes with an average of ≥5 BMCs in one or more study groups were considered. Shown here are CYP and detoxification genes with relevance to BaP metabolism and conjugation reactions. Please see text for further details.

are the expression of AHR and ARNT (Aryl-Hydrocarbon Receptor and Aryl-Hydrocarbon Nuclear Translocator, respectively) involved in binding of BaP and other PAH, important for transcriptional activation of genes involved in BaP activation and detoxification.

## Discussion

This study shows that altering GSH levels affects BaP adducts levels both within and outside T47D cells. The amount of BaP adducts was almost five to ten times greater in the extracellular vs cellular space despite substantial cellular GSH levels and BaP conjugate formation. The time-dependent formation of cellular and extracellular protein adducts provided insight into the underlying processing of BaP. BSO lowered GSH and in turn cellular protein adducts rose to a maximum specific adduct binding after 18 hr at about 10-fold higher than extracellular protein specific adduct binding. However, the total amount of extracellular adducts continued to climb over 48 hr. These data suggest extracellular protein in media provides accessible binding sites for BaP reactive metabolites if cellular sites become saturated, or if conjugation systems are simply evaded by lipid-stabilized diffusion [51] of reactive BaP metabolites diffusing through the intracellular matrix and into the extracellular space. The stability of BPDE to cross cell membranes and form DNA/protein adducts has been documented both *in vitro* in cell culture [52] and *in vivo* after intraperitoneal injection in mice where DNA adducts were found in distal tissues (e.g., lung, liver, kidney) [40]. We surmised in our study that: 1) the higher specific adduct binding of cellular protein adducts was due to their proximity of reactive metabolites as they were being formed by CYP metabolism; and 2) excess reactive BaP metabolites such as BPDE that escaped conjugation were sufficiently stable to diffuse into the extracellular space to encounter protein binding sites on FCS proteins. In addition to BaP metabolite diffusion, a possible contribution of efflux transporters like ABCG2 [53] in moving BaP metabolites and reactive intermediates across cell membranes in mammary epithelia deserves inquiry. Further, the formation of BaP protein adducts under conditions of varying FCS levels including deprivation, BSA depleted FCS, and dose-dependency of BaP treatment, awaits further investigation.

T47D cells are well equipped for generating reactive BaP metabolites that result in DNA or protein binding. Recent RNA-seq data [48–50] (Fig 6) agree with prior experimental studies [54,55] in T47D cells on the prevalence CYP1B1 with contributions from CYP1A1 as primary enzymes for BaP bioactivation. For detoxification reactions, RNA-seq data also revealed why BaP sulfate conjugates prevail over glucuronides due to expression of four sulfotransferases, especially SULT1A1 as previously reported [56] and as we report now, also SULT2B1, over the moderate expression of only one glucuronyltransferase (UGT3A2). In our study, we found GSH-S transferase (GST) enzyme activity to the model substrate CDNB was robust, (>50 nmole product/min/mg protein) reflecting the activity of multiple GST genes in T47D cells. Generally, GST enzyme activity occurs primarily in cytosol, although nuclear localization has occasionally been observed [57]. GSH conjugations to BPDE have been documented *in vitro* for GSTA1, GSTA2, GSTM1, GSTM2, and GSTP1 [52,58–64]; and in epidemiology studies for GSTA1 (GST-alpha), GSTM1, GSTP1, GSTT1, and GSTT2 (GST-theta) [65–68]. Of these GST genes, T47D cells express GSTM2, GSTP1, GSTT2 but other expressed GST's may also play a role in BaP conjugate formation. Further, inactivation of BaP reactive metabolites is complicated by the presence of multiple GST genetic variants in the human population with different binding affinities for BPDE [58,69]. Nevertheless, we found T47D cells produce a sizeable proportion of water soluble metabolites at 35–36% (Fig 3) after removal of other conjugates (sulfate, glucuronide) and any free hydroxylated metabolites, suggesting these are GSH conjugates. Future studies including western blot and qPCR analyses could clarify these findings from RNA-seq data.

An inverse relationship between [3H]-BaP adducts and cellular GSH levels was clearly evident for 1) DNA/nuclear protein adducts and 2) cellular protein and total [3H]-BaP adducts formed during the first 24 hr of incubation. BSO depletion of GSH had the greatest effect on both specific adduct binding and total protein adduct formation by depriving GST enzymes of GSH as a required conjugation substrate for BaP electrophiles. Nuclear protein adducts doubled and DNA adducts increased by 29%. Altered DNA repair by BaP metabolites can affect DNA adduct levels [70] but such effects may be limited here by short incubation times. By using BaP pretreatment to increase GSH, we recognize a more complex scenario

occurs from inductive effects of AHR-mediated transcription but we were interested in effects of repeated BaP exposure. We observed that the ameliorating effects of increased GSH and GST activity were relatively minor for DNA and nuclear protein binding but were significant over 24 hr in lowering the specific adduct binding of cellular protein adducts (Fig 2 Panel A) and decreasing total protein adducts (Fig 2 Panel D). However, even with an excess supply of cellular GSH above control levels, substantial adduct formation still occurred. These results suggest a chemoprevention strategy of simply increasing cellular GSH alone without accompanying elevations in GSTs or other enzymes would be ineffective for reducing tumor incidence from chemical exposures [71].

In our study, BaP reactive metabolites (e.g., epoxides) that escaped GSH conjugation were sufficiently stable to migrate into the extracellular space and bind to albumin and α1AT from fetal calf serum. BaP-BSA (bovine serum albumin) adducts are well known and can be formed by direct reaction of BPDE with purified BSA *in vitro* [72–74]. To our knowledge ours is the first report of BaP-α1AT adducts. Extensive analysis of human and mouse serum albumin has shown BPDE adducts occur at histidine and lysine residues [75,76]. Fetal calf serum is prepared for cell culture to minimize the presence of immunoglobulins and coagulation-related proteins so that albumin and α1AT are highly abundant potential targets for BaP reactive metabolites. Although we do not know the exact amino acid binding sites for BaP in these two proteins, sequence conservation of albumin and α-1AT among species makes histidine and lysine residues as likely sites for BaP adduct formation. Indeed, that α-1AT is a target for BaP adduct formation is suggested by reports of other chemicals targeting α-1AT to form adducts that includes malondialdehyde (A1AT[284-298]) at H293 [77], and also acrolein, binding at lysine and histidine residues [78]. While we do not discount leakage of some cellular BaP adducts into media, our data suggests BSA and α1AT make up most of the extracellular adducts.

Cellular protein adducts were also separated by 2D-PAGE in this study. The radiometric pattern of BaP adducts detected did not correspond to protein abundance indicated by silver staining. This should not be unexpected since only a discrete subpopulation of proteins in their native 3D configuration would have a limited number of specific residues (e.g., unmodified His, Lys amino acids) externalized for BaP reactive metabolite binding. Over 100 $^3$H-BaP adducted proteins were observed and we speculate many more BaP adducts were unaccounted for due to limitations of fluorographic detection. Identification of BaP-adducted proteins and the specific adducted residues in T47D cells would be of great interest, especially since the 2D-gel pattern of protein adducts was essentially unchanged by GSH depletion (S2 Fig).

Some limitations to this study should be mentioned. In keeping with our prior work, we conducted current studies at a fixed concentration of 4 µM BaP. Incubation with lower BaP concentrations at environmental exposure levels would likely reduce DNA and protein adduct formation but not eliminate them entirely. Many epidemiological studies find BPDE-DNA and protein adducts in a diversity of human populations and living conditions [30,79,80], indicating the ubiquitous nature of adduct formation under long term, low level, environmental exposures. Another point is i*n vitro* incubations do not mimic physiologic conditions including a 3D-architecture, the presence of mixed cell types (e.g., adipocytes, fibroblasts, endothelial and immune cells) and blood flow to supply nutrients and remove BaP metabolites. Finally, T47D cells are a cancer cell line notably expressing AHR that activates CYP1A1 and CYP1B1 expression and many downstream genes frequently found in breast cancers [81]. Despite these limitations, researchers have shown BaP can be efficiently metabolized by normal mammary epithelial cells (similar to T47D cells) which results in DNA binding [82] and mutation patterns in cancer driver genes characteristic of human tumors [83]. As such, cell lines like T47D cells continue to provide value in studying breast cancer etiology and intervention research [84].

## Conclusion

Extracellular protein adducts accounted for almost 10% of BaP metabolized by T47D cells. Serum albumin and α1AT were primary targets resulting in BaP adducts. These results suggest BaP reactive metabolites like BPDE can easily translocate across cell membranes despite ample conjugation systems and an available supply of essential co-substrates for sulfate or GSH-mediated enzymes. The implications *in vivo* are that BaP reactive metabolites could enter adjacent epithelia with

some fraction resulting in DNA binding and potentially somatic mutations in cancer susceptibility genes over time [85]. Associations continue to grow for PAH exposure and pollution as important factors affecting human health, including their potential involvement in mammary epithelial carcinogenesis [86,87]. The multigenic origins of a heterogeneous disease like breast cancer highlight the need for a broader understanding of gene-environment interactions over the developmental stages of mammary tissues throughout the human life span [88].

## Supporting information

**S1 Fig. Metabolic pathway formation of BaP adducts or GSH-conjugates.** Metabolic pathway for BaP adduct formation of macromolecules is shown. Other BaP electrophiles may be involved in adduct formation as well. Enzymatic GSH conjugation of BaP is a detoxification pathway.
(TIF)

**S2 Fig. ³H-BaP cellular protein adducts after DMSO, BSO or BaP.** T47D cells were pretreated for 48 hr with 0.5 mM BSO or 4 μM BaP (unlabeled) to decrease or increase GSH compared to DMSO control (0.4 v/v). Pretreated cells were then exposed to 4 μM ³H-BaP (6 Ci/mmole) for 24hr after which cellular protein was collected, acetone precipitated, washed to remove unbound radioactivity, lyophilized and then solubilized according to methods. A 500,000 dpm amount of labeled protein was separated by 2D PAGE, after which gels were fixed, dried and exposed to film for 8 weeks at −80°C.
(TIF)

**S1 Table. BaP metabolite distribution in T47D cells after 24 hr incubation.**
(XLSX)

**S2 Table. RNA-seq data from 3 independent studies, aligned to hg38.**
(XLSX)

**S1 Raw Images. Raw gel images with descriptive text are contained in the supporting information file.**
(PDF)

## Acknowledgments

We gratefully acknowledge Dr Eric J. Tokar (MTB) and Dr Birandra Sinha (MTB) for internal reviews of the manuscript. We greatly appreciate leadership support of this research by the Branch Chief, Dr Darlene Dixon, and by the DTT Scientific Director, Dr Heather B. Patisaul. Logistical support by DTT Administrative Staff are gratefully acknowledged.

## Author contributions

**Conceptualization:** B. Alex Merrick.

**Data curation:** B. Alex Merrick, Ashley M. Brooks.

**Formal analysis:** B. Alex Merrick, Ashley M. Brooks.

**Funding acquisition:** B. Alex Merrick, James K. Selkirk.

**Investigation:** B. Alex Merrick, Betty K. Mansfield.

**Methodology:** B. Alex Merrick, Ashley M. Brooks, Betty K. Mansfield, James K. Selkirk.

**Project administration:** B. Alex Merrick, James K. Selkirk.

**Resources:** James K. Selkirk.

**Software:** Betty K. Mansfield.

**Supervision:** B. Alex Merrick, James K. Selkirk.

**Writing – original draft:** B. Alex Merrick.

**Writing – review & editing:** B. Alex Merrick, Ashley M. Brooks, Julie F. Foley, Betty K. Mansfield, James K. Selkirk.

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
