## [Decision Letter · Decision Letter 0]

27 Sep 2025

Dear Dr. Merrick,

Thank you for submitting your manuscript to PLOS ONE. After careful consideration, we feel that it has merit but does not fully meet PLOS ONE’s publication criteria as it currently stands. Therefore, we invite you to submit a revised version of the manuscript that addresses the points raised during the review process.

In the submitted manuscript, authors have aimed to evaluate the translocation of Benzo(a)pyrene (BaP) reactive metabolites across human mammary epithelial cell membranes and the findings of this manuscript suggest that BaP reactive metabolites like BPDE (BaP-diol-epoxide) can easily translocate across cell membranes despite robust conjugation systems and ready supplies of essential co-substrates for sulfate or GSH conjugations. However, the submitted manuscript needs major revision and cannot be recommended in its current form for publication in esteemed journal Plos One. It is requested that authors must revise the manuscript according to learned reviewers’ suggestions. The corrections made in the manuscript should be highlighted. So, it would be easier to identify the modified content from the original submitted manuscript. 

We look forward to receiving your revised manuscript.

Kind regards,

Pankaj Singh, Ph.D.

Academic Editor

PLOS ONE

Journal Requirements:

2. Thank you for stating the following financial disclosure: [This research was internally funded by the Intramural Research Program of the NIH, National Institute of Environmental Health Sciences, through ZIA ES103378 from the Division of Translational Toxicology.]. 

Reviewers' comments:

Reviewer's Responses to Questions

**Comments to the Author**

1. Is the manuscript technically sound, and do the data support the conclusions?

Reviewer #1: Partly

Reviewer #2: Yes

Reviewer #3: Yes

2. Has the statistical analysis been performed appropriately and rigorously?

Reviewer #1: N/A

Reviewer #2: I Don't Know

Reviewer #3: Yes

3. Have the authors made all data underlying the findings in their manuscript fully available?

Reviewer #1: Yes

Reviewer #2: Yes

Reviewer #3: Yes

4. Is the manuscript presented in an intelligible fashion and written in standard English?

Reviewer #1: Yes

Reviewer #2: Yes

Reviewer #3: Yes

Reviewer #1: Breast cancer is the most common type of cancer in women, and the global incidence of this disease continues to rise.

Despite advancements in early detection and therapeutic strategies, breast cancer remains a remarkable health challenge because of its complex pathogenesis and diverse clinical manifestations.

Like all kinds of cancer, the etiology of breast cancer involves a variety of factors, including genetics, diet, lifestyle, and environmental factors. Nonetheless, germline mutations predisposing to breast cancer account for 10% of the cases, and the mutated genes encode for proteins involved in the DNA-damage repair, cell-cycle control, and chromatin remodeling processes. On the other hand, somatic mutations are the most frequent. A significant contribution to somatic mutations is due to environmental exposure. Within this framework, a considerable contribution to the mutation burden is due to polycyclic aromatic hydrocarbons (PAHs), of which benzo (a) pyrene (BaP) is a representative. PHAs' exposure occurs through multiple sources, including natural (e.g., volcanic eruption), industrial (e.g., air pollution due to industrial emissions and vehicle exhausts), and lifestyle choices (e.g., tobacco smoke). BaP is a potent DNA mutagenic molecule, thus tightly associated with carcinogenesis. However, cells counteract such "xenobiotic" stress caused by BaP exposure by creating less toxic BaP adducts, also thanks to the intervention of glutathione (GSH). Based on their previous inquiries in the manuscript titled "Translocation of Benzo(a)pyrene Reactive Metabolites Across Human Mammary Epithelial Cell Membranes", Merrick A.B. and colleagues tested the hypothesis of a potential inverse distribution of cellular DNA/protein adducts and extracellular protein BaP upon GSH depletion or augmentation in T47D cells. Overall, the manuscript presents some interesting hints; nonetheless, before publication amendments are required.

* Given that there is no mention of any form of a specific enzyme kinetic rate in the entire article, I would exercise caution when using the phrase "…specific activity…" as it could be misunderstood or cause confusion for the readers. The rate at which a substrate is converted into a particular product by unit time is sometimes referred to as specific activity.

* More solid evidence regarding the greater quantity of extracellular protein adducts (such as BSA and alpha 1 anti-trypsin) in comparison to intracellular protein is requested from the authors. Since BSA is the main protein component and a primary part of FBS, I must admit that I am not startled if we consider that cells are cultivated in the presence of FBS. What happens if cells are starved? Or reducing the amount of serum? Does a dosage dependency exist? BSA depleted FBS?

* Robust mechanistic data supporting the translocation of BaP reactive metabolites across a phospholipid bilayer are missing. I mean, while BaP entering cells is easy due to the very hydrophobic features of the molecules, once the molecule has been hydroxylated, its features change. Indeed, hydroxylated forms are more polar, thus cannot freely diffuse through the plasma membrane as effectively as the parent compound. A possibility is that both adducts and BaP-free hydroxylated forms are secreted via the Endoplasmic Reticulum (ER). Consistently, CYP proteins, which are mostly localized within the ER, and GSH coordinate detoxification through sequential action where first CYP activate, or functionalize the BaP, making it more water soluble, and then GSH can conjugate with these modified substrates, rendering them less toxic and facilitating their secretion. The authors should clarify this issue. Did they ever check ER? To gain insight into the issue, this route should be considered.

* Ultimately, the CYP issue needs to be strengthened. The authors refer only to the transcript amount, and even from cells cultured in different conditions. As such, this part is a little bit too speculative. I would advise either strengthening it by providing protein data (e.g., Western Blot) or omitting this last part.

Minor concerns

* Acronyms and abbreviations when cited for the first time should be spelled out (e.g., line 126: CNDB; line 547: FCS; etc.)

* A few typos seem scattered through the main text (e.g., line 111, etc.).

Reviewer #2: This is a thoughtful, well designed study on an important biological question. The manuscript is well written and easy to understand even for people outside the field. The basic scientific question asked is simple, which I consider a plus. The work does not provide answers to how these adducts affect human health, but the information provided here will be important for future studies.

Reviewer #3: The authors investigated the dynamics of BaP adduct formation and distribution in T47D mammary cells in connection with the role of BaP and PAH in tumor formation and involvement in cancer malignancies. The authors showed the kinetics of the distribution of BaP adducts by using BSO and Bap treatments/pre-treatments to alter GSH levels in the cells. The authors also mined the available datasets to demonstrate the expression levels of the enzymes and transcription factors, relevant for Bap conjugation and bioactivation.

The data is sound, generally well interpreted. Methods are presented with appropriate references where applicable. Legends contain necessary details.

Figure 1 and 2 can be merged into 2 panel figure. Same for Figure 3 and 4 – 4 panel single figure.

Lines 332 -334 “…….but only decreased nuclear protein adducts by 11% from control”. Table 1 however indicates an increase in nuclear protein adducts by 11%. The authors should correct and elaborate.

While the authors provide plausible explanation for the observed kinetics differences in Figure 3, how would the authors explain the difference in BSO curve relative to control in cellular vs extracellular adduct formation? Is it expected that GSH activity induced by BSO treatment would lead to lower escape to the media? The trend in total BaP adduct in Fig 4 goes more in line with the cellular BaP adduct kinetics of Fig 3. Please, include a sentence either in results or discussion.

For RNA-seq analysis the authors should cite packages used for processing appropriately and move the dataset accession numbers from Methods to Data Availability section.

The authors should give the article additional round of proofreading and address these minor corrections.

**Do you want your identity to be public for this peer review?** For information about this choice, including consent withdrawal, please see our Privacy Policy

Reviewer #1: No

Reviewer #2: No

Reviewer #3: No

---

## [Author Response · Author response to Decision Letter 1]

20 Oct 2025

Date:October 18, 2025

To:Pankaj Singh, Ph.D.

Academic Editor

PLOS ONE

From: B. Alex Merrick, Ph.D.

NIEHS/NIH

Subject: Revision of PONE-D-25-43097

Dear Dr Singh,

Thank you for your decision to REVISE our manuscript, “Translocation of benzo(a)pyrene reactive metabolites across human mammary epithelial cell membranes”, PONE-D-25-43097. Reviewer comments helped us improve our manuscript. We are providing the following requested information.

1) Rebuttal Letter: In the current rebuttal letter submitted as a file labeled, “Response to Reviewers”, I enclose the changes to meet Journal requirements and a point-by point response to each Reviewer’s comment.

2) Marked-up Revision: A marked-up version of the original manuscript is submitted as a separate file, “Revised Manuscript with Track Changes”.

3) Unmarked Revision: An unmarked version of the revised manuscript is submitted as “Manuscript”.

4) Revised Figures, Supplemental data and Raw Gel File: Figures 1 & 2 were merged and Figures 3 &4 were merged into single figures, and were re-submitted as Fig 1 and Fig 2, respectively. Figure 5 (formerly Fig 7) was revised to comply with the Raw gel requirements while still showing the same 2D gel protein separations as before. All manuscript figures were re-numbered accordingly in the manuscript text.

5) Raw Gel Images in Supplemental Information file: Raw, uncropped, original gel images according to journal guidelines were combined into one PDF file with descriptive text. The file was labeled, “S1_Raw_Images” as a Supporting Information file.

Requested Journal Requirements:

1. Title Page: The title page has been revised to fit Plos One style requirements.

2. Funding Statement: The funding statement has been revised to the following: “This research was internally funded by the Intramural Research Program of the NIH, National Institute of Environmental Health Sciences, through ZIA ES103378 from the Division of Translational Toxicology. The funders had no role in study design, data collection and analysis, decision to publish, or preparation of the manuscript.”

3. Raw gel images – Raw, uncropped gel images have been provided with descriptive text in a single, PDF file labeled, “S1_Raw_Images”.

4. Plos One Format: The manuscript title has been changed to meet Plos One format so the online submission has been corrected.

5. Ethics Statement: The Ethics Statement has been added to the Methods section with the following text. “The authors obtained oral consent to use a human cell line from the supervisory Branch Chief at NIEHS. Use of biospecimens obtained from an established tissue repository are exempt from NIEHS IRB review. All cells were disposed of responsibly when experiments were completed. The authors have no additional ethical concerns for this study.”

6. Citation: Reviewer #1 rightly refers to one of our previous publications (PMID: 4075284) which is relevant to the current study, so it should be cited.

Responses to Reviewer Comments

REVIEWER #1:

COMMENT: Breast cancer is the most common type of cancer in women, and the global incidence of this disease continues to rise.

Despite advancements in early detection and therapeutic strategies, breast cancer remains a remarkable health challenge because of its complex pathogenesis and diverse clinical manifestations.

Like all kinds of cancer, the etiology of breast cancer involves a variety of factors, including genetics, diet, lifestyle, and environmental factors. Nonetheless, germline mutations predisposing to breast cancer account for 10% of the cases, and the mutated genes encode for proteins involved in the DNA-damage repair, cell-cycle control, and chromatin remodeling processes. On the other hand, somatic mutations are the most frequent. A significant contribution to somatic mutations is due to environmental exposure. Within this framework, a considerable contribution to the mutation burden is due to polycyclic aromatic hydrocarbons (PAHs), of which benzo (a) pyrene (BaP) is a representative. PHAs' exposure occurs through multiple sources, including natural (e.g., volcanic eruption), industrial (e.g., air pollution due to industrial emissions and vehicle exhausts), and lifestyle choices (e.g., tobacco smoke). BaP is a potent DNA mutagenic molecule, thus tightly associated with carcinogenesis. However, cells counteract such "xenobiotic" stress caused by BaP exposure by creating less toxic BaP adducts, also thanks to the intervention of glutathione (GSH). Based on their previous inquiries in the manuscript titled "Translocation of Benzo(a)pyrene Reactive Metabolites Across Human Mammary Epithelial Cell Membranes", Merrick A.B. and colleagues tested the hypothesis of a potential inverse distribution of cellular DNA/protein adducts and extracellular protein BaP upon GSH depletion or augmentation in T47D cells. Overall, the manuscript presents some interesting hints; nonetheless, before publication amendments are required.

RESPONSE: Reviewer#1 provides a very nice summary of the breast cancer field and the need for continued studies on the etiology of somatic mutations in breast malignancies and other cancer types.

COMMENT: Given that there is no mention of any form of a specific enzyme kinetic rate in the entire article, I would exercise caution when using the phrase "…specific activity…" as it could be misunderstood or cause confusion for the readers. The rate at which a substrate is converted into a particular product by unit time is sometimes referred to as specific activity.

RESPONSE: Reviewer#1 makes an excellent point about possible confusion with enzyme kinetics regarding the phrase, “specific activity”. This phrase has been changed to “specific adduct binding” throughout the manuscript and has been defined in subsection BaP Adduct Measurements in Methods as follows: “The term ‘specific adduct binding’ refers to the amount of adducts specific per mg protein (e.g. pmole BaP adducts/mg protein) or per mg DNA, so as to normalize adduct formation according to protein levels in cells or culture media (extracellular protein) or DNA levels.”

COMMENT: More solid evidence regarding the greater quantity of extracellular protein adducts (such as BSA and alpha 1 anti-trypsin) in comparison to intracellular protein is requested from the authors. Since BSA is the main protein component and a primary part of FBS, I must admit that I am not startled if we consider that cells are cultivated in the presence of FBS. What happens if cells are starved? Or reducing the amount of serum? Does a dosage dependency exist? BSA depleted FBS?

RESPONSE: We feel confident that the radioactive binding to protein precipitates of culture media protein (including BSA, alpha-1AT and others) is good evidence of greater binding compared to cellular protein BaP adducts. However, Reviewer#1 raises intriguing questions about BaP adducts during conditions of cell starvation, serum reduction, dose dependency, and BSA depletion from FBS. These are great opportunities for future study, as highlighted in the Discussion by the following text.

“Further, the formation of BaP protein adducts under conditions of varying FCS levels including deprivation, BSA depleted FCS, and dose-dependency of BaP treatment, awaits further investigation.”

COMMENT: Robust mechanistic data supporting the translocation of BaP reactive metabolites across a phospholipid bilayer are missing. I mean, while BaP entering cells is easy due to the very hydrophobic features of the molecules, once the molecule has been hydroxylated, its features change. Indeed, hydroxylated forms are more polar, thus cannot freely diffuse through the plasma membrane as effectively as the parent compound. A possibility is that both adducts and BaP-free hydroxylated forms are secreted via the Endoplasmic Reticulum (ER). Consistently, CYP proteins, which are mostly localized within the ER, and GSH coordinate detoxification through sequential action where first CYP activate, or functionalize the BaP, making it more water soluble, and then GSH can conjugate with these modified substrates, rendering them less toxic and facilitating their secretion. The authors should clarify this issue. Did they ever check ER? To gain insight into the issue, this route should be considered.

RESPONSE: Reviewer#1 raises several interesting questions about the mechanism of translocation of reactive metabolites. The general consensus in the field is that hydroxylated and conjugated BaP metabolites, even though more polar than BaP itself, are still very hydrophobic and passively diffuse through cytoplasm and across the cellular membrane (Dock et al 1987, PMID 3102083). Reactive epoxide metabolites like BPDE can either hydrolyze to tetrols or bind to macromolecules such as intracellular protein, nuclear protein, nuclear DNA, or as we have found, bind to extracellular protein in the culture media. While passive diffusion of BPDE is a probable mechanism, it is still possible that efflux transporters like ABCG2 may play a mechanistic role in BaP transport across cellular membranes into the extracellular space. This possibility is now placed in the Discussion section in the following sentence.

“In addition to BaP metabolite diffusion, a possible contribution of efflux transporters like ABCG2 [53] in moving BaP metabolites and reactive intermediates across cell membranes in mammary epithelia deserves inquiry.”

Also, in answer to Reviewer#1’s query about the role of ER, we did not examine ER secretory activity or other subcellular organelles in this study, but this would be a fascinating area to explore in future work.

COMMENT: Ultimately, the CYP issue needs to be strengthened. The authors refer only to the transcript amount, and even from cells cultured in different conditions. As such, this part is a little bit too speculative. I would advise either strengthening it by providing protein data (e.g., Western Blot) or omitting this last part.

RESPONSE: The expression of CYP1A1 and CYP1B1 in T47D cells has been previously reported by Kim et al 1998 and Spink et al 1998, as we have originally referenced in the text. The induction of CYP1A1 and CYP1B1 by BaP and other polyaromatic hydrocarbons is well known. For us, we felt there was added value in analyzing and comparing RNA-seq data from the control conditions of three different T47D studies. In doing so, we gained a wider perspective on all expressed CYPs as well as other relevant detoxification genes in T47D cells. Future studies including western blot and qPCR analyses could further our work. In response, we have added the following sentence to the Discussion.

“Future studies including western blot and qPCR analyses could further clarify these findings from RNA-seq data.”

Minor concerns

COMMENT: Acronyms and abbreviations when cited for the first time should be spelled out (e.g., line 126: CNDB; line 547: FCS; etc.)

RESPONSE: In addition to our list of abbreviations, we have spelled the full meaning of acronyms and abbreviations in the text when used the first time.

COMMENT: A few typos seem scattered through the main text (e.g., line 111, etc.).

RESPONSE: We thank Reviewer#1 for this comment and have endeavored to find and correct any typos in the text.

REVIEWER #2:

COMMENT: This is a thoughtful, well designed study on an important biological question. The manuscript is well written and easy to understand even for people outside the field. The basic scientific question asked is simple, which I consider a plus. The work does not provide answers to how these adducts affect human health, but the information provided here will be important for future studies.

RESPONSE: We thank Reviewer#2 for these positive comments and the question about how these adducts affect human health. BaP or other polyaromatic hydrocarbons are known carcinogens and their adducts are indicators of exposure, but there is not yet a clear linkage to health effects like breast cancer. We have modified the following sentence in the Conclusion to reflect this point. “Associations continue to grow for PAH exposure and pollution as important factors affecting human health, including their potential involvement in mammary epithelial carcinogenesis.”

REVIEWER #3

COMMENT: The authors investigated the dynamics of BaP adduct formation and distribution in T47D mammary cells in connection with the role of BaP and PAH in tumor formation and involvement in cancer malignancies. The authors showed the kinetics of the distribution of BaP adducts by using BSO and Bap treatments/pre-treatments to alter GSH levels in the cells. The authors also mined the available datasets to demonstrate the expression levels of the enzymes and transcription factors, relevant for Bap conjugation and bioactivation.

The data is sound, generally well interpreted. Methods are presented with appropriate references where applicable. Legends contain necessary details.

RESPONSE: We thank Reviewer#3 for these positive comments and our efforts to mine published T47D datasets to demonstrate expression levels of enzymes and transcription factors relevant for BaP metabolism and detoxification.

COMMENT: Figure 1 and 2 can be merged into 2 panel figure. Same for Figure 3 and 4 – 4 panel single figure.

RESPONSE: We have merged Figures 1 and 2 into a two-panel single figure (now called Figure 1 Panels A and B). We have also merged Figures 3 and 4 into a single figure (now called Figure 2 Panels A, B, C and D). Figure legends have been revised accordingly to reflect these mergers and the Results text referring to the new figures numbers and panels has also been updated.

COMMENT: Lines 332 -334 “…….but only decreased nuclear protein adducts by 11% from control”. Table 1 however indicates an increase in nuclear protein adducts by 11%. The authors should correct and elaborate.

RESPONSE: We thank Reviewer#3 for bringing this discrepancy to our attention. We have corrected this description (bold font) with the following text in the Results section. (Note that revised Figure numbers/panels are used in the new manuscript version.)

“The effects of GSH depletion or induction upon BaP adduct formation were also determined in T47D cell nuclei. After near depletion of GSH with BSO (Table 1), DNA adducts/mg increased by 29% after 24 hr while nuclear protein adducts/mg almost doubled. Increasing GSH by BaP pretreatment did significantly lower DNA adducts/mg by 13% but did not significantly affect nuclear protein adducts/mg. Interestingly, when protein BaP adduct formation from whole cell lysates was followed over time (Fig 2, Panel A), it generally followed the same proportional binding pattern as nuclear protein adducts for BSO treatment. At 18 and 24 hr, specific protein adduct binding in cells after GSH depletion was almost doubled from the DMSO control. However, GSH increases from BaP pretreatment lowered cellular adducts in Fig 2, Panel A by relatively lesser amounts to 20-25% over 12 to 24 hr from DMSO control, while nuclear protein binding with BaP pretreatment (Table 1) was not statistically lowered from control. Apparently, the increase in GSH caused by BaP pretreatment was not substantial enough to lower nuclear protein binding.”

COMMENT: While the authors provide plausible explanation for the observed kinetics differences in Figure 3, how would the authors explain the difference in BSO curve relative to control in cellular vs extracellular adduct formation? Is it expected that GSH activity induced by BSO treatment would lead to lower escape to the media? The trend in total BaP adduct in Fig 4 goes more in line with the cellular BaP adduct kinetics of Fig 3. Please, include a sentence either in results or discussion.

RESPONSE: Reviewer#3 raises a good point about more thoroughly explaining how adduct formation was affected by lower GSH levels from BSO in Figures 3 and 4 – now merged into Figure 2. The following paragraph has been added to the Results.

“Data in Fig 2 demonstrate the kinetics of protein adduct formation is primarily influenced by 1) the amount of cellular vs extracellular protein available for binding and 2) the proximity to reactive metabolite formation within the cell. In Fig 2

---

## [Decision Letter · Decision Letter 1]

10 Nov 2025

Translocation of benzo(a)pyrene reactive metabolites across human mammary epithelial cell membranes

PONE-D-25-43097R1

Dear Dr. Merrick,

We’re pleased to inform you that your manuscript has been judged scientifically suitable for publication and will be formally accepted for publication once it meets all outstanding technical requirements.

Kind regards,

Pankaj Singh, Ph.D.

Academic Editor

PLOS ONE

Additional Editor Comments (optional):

Reviewers' comments:

Reviewer's Responses to Questions

**Comments to the Author**

Reviewer #2: All comments have been addressed

Reviewer #3: All comments have been addressed

2. Is the manuscript technically sound, and do the data support the conclusions?

Reviewer #2: Yes

Reviewer #3: Yes

3. Has the statistical analysis been performed appropriately and rigorously?

Reviewer #2: I Don't Know

Reviewer #3: Yes

4. Have the authors made all data underlying the findings in their manuscript fully available?

Reviewer #2: Yes

Reviewer #3: Yes

5. Is the manuscript presented in an intelligible fashion and written in standard English?

Reviewer #2: Yes

Reviewer #3: Yes

Reviewer #2: The authors have directly addressed my concern that while their work is important, well written and technically sound, it does not answer the question of how these adducts affect carcinogenesis and human health; This concern has been adequately addressed by the authors in the Conclusion. I want to reiterate, however, that I have no doubt that BaP adducts cause cancer, but the cell and molecular mechanisms for this carcinogenesis remains unknown. I hope the authors continue this important work.

Reviewer #3: Line 377 the word "reactive" is double typed, please check. The rest of the concerns are corrected and addressed.

**Do you want your identity to be public for this peer review?** For information about this choice, including consent withdrawal, please see our Privacy Policy

Reviewer #2: No

Reviewer #3: No

---

## [Editor Report · Acceptance letter]

PONE-D-25-43097R1

PLOS ONE

Dear Dr. Merrick,

I'm pleased to inform you that your manuscript has been deemed suitable for publication in PLOS ONE. Congratulations! Your manuscript is now being handed over to our production team.

Kind regards,

on behalf of

Dr. Pankaj Singh

Academic Editor

PLOS ONE